# Spreading Dynamics of Capital Flow Transfer in Complex Financial Networks

**DOI:** 10.3390/e25081240

**Published:** 2023-08-21

**Authors:** Wenyan Peng, Tingting Chen, Bo Zheng, Xiongfei Jiang

**Affiliations:** 1Department of Physics, Zhejiang University, Hangzhou 310018, China; 11636017@zju.edu.cn; 2Department of Finance, Zhejiang University of Finance and Economics, Hangzhou 310018, China; 3School of Physics and Astronomy, Yunnan University, Kunming 650091, China; 4College of Finance and Information, Ningbo University of Finance and Economics, Ningbo 315175, China; jiangxiongfei@nbufe.edu.cn

**Keywords:** complex networks, financial systems, capital flow, transfer entropy, econophysics

## Abstract

The financial system, a complex network, operates primarily through the exchange of capital, where the role of information is critical. This study utilizes the transfer entropy method to examine the strength and direction of information flow among different capital flow time series and investigate the community structure within the transfer networks. Moreover, the spreading dynamics of the capital flow transfer networks are observed, and the importance and traveling time of each node are explored. The results imply a dominant role for the food and drink industry within the Chinese market, with increased attention towards the computer industry starting in 2014. The community structure of the capital flow transfer networks significantly differs from those constructed from stock prices, with the main sector predominantly encompassing industry leaders favored by primary funds with robust capital flow connections. The average traveling time from sectors such as food and drink, coal, and utilities to other sectors is the shortest, and the dynamic flow between these sectors displays a significant role. These findings highlight that comprehension of information flow and community structure within the financial system can offer valuable insights into market dynamics and help to identify key sectors and companies.

## 1. Introduction

The financial system, a complex yet crucial network, facilitates capital exchange, with information serving a vital function, while the majority of research on capital flows concentrates on the characteristics and influencing factors of inter-country capital flows, various analytical methods have been employed. Commonly utilized approaches such as regression and modeling analysis offer insights into capital flow. For instance, regression analysis assesses the effect of global financial cycles (the COVID shock) and uncertainties of cross-border capital flow [1,2,3], with studies indicating that heightened capital flow uncertainty often triggers market perturbations and crashes [4]. Additionally, models have been designed to examine the impact of production systems on capital flows [5]. A growing trend focuses on the analysis of capital flows through the complex network theory, which offers a holistic perspective on the intricate interactions within capital flow networks. For instance, researchers have scrutinized the global capital network by assessing cross-border capital flows across 65 countries [6]. Complex network methods have also been adopted to explore the spillover effects of foreign capital between stock markets and other financial assets [7]. Furthermore, the construction and examination of illicit capital flow networks using omission measurements have been carried out [8]. Despite these advancements, understanding of the intricate interactions within a company’s capital flow network remains limited.

In recent years, high order dependencies analysis has been applied to go beyond the pairwise description of markets [9,10]. As an example of such high order dependencies analysis, the use of entropy-based methods in the financial system has gained considerable interest among researchers recently [11,12,13,14,15]. Transfer entropy (TE), a technique derived from the concept of entropy, is utilized to investigate the information flow within the financial system. Previous studies have employed TE successfully to construct and analyze information transfer networks based on various financial assets, including stock prices and cryptocurrencies such as Bitcoin [16,17,18,19]. By leveraging TE, researchers have estimated the influence of one time series’ historical behavior on another. Complex financial systems often exhibit characteristic community structures [20,21,22,23,24,25], wherein stocks within a community fluctuate simultaneously. Recognizing this feature, this study uses the transfer entropy method to extract the strength and direction of information flow between different capital flow time series. It also delves into the analysis of the community structure present within the transfer networks.

The current trend in complex network studies aims to integrate network topology and dynamics, in contrast to earlier works that focused solely on either network topology [16,26,27,28] or dynamics [29,30,31,32,33,34,35,36]. To bridge the gap between the two aspects, this approach facilitates the tracking of information flow spreading dynamics in complex networks, highlighting the importance of specific nodes in capital propagation and the speed of signal transmission [37,38,39]. By drawing a parallel between financial information spread and infectious diseases, epidemic models have been widely applied in financial system studies [30,34,35]. With this in mind, our study incorporates the Monte Carlo simulation of the susceptible–infected–susceptible (SIS) infectious disease model into the capital flow transfer network, thereby exploring the significance and traveling time of each node within complex financial networks.

This paper is structured as follows: Section 2 provides an in-depth overview of the data and methodologies used. In Section 3, we construct the capital flow transfer network and analyze its community structure. Additionally, we observe the spreading dynamics of the capital flow transfer networks by studying the importance and duration of each node. Finally, in Section 4, we discuss and draw conclusions based on our findings.

## 2. Materials and Methods

### Data

The primary dataset employed in this research comprises capital flow data denoted as $CF_{i}\left(t\right)$ for the constituents of the HS300 index in the Chinese stock market, ranging from January 2010 to December 2022. The term “capital flow” denotes the volume of capital flowing in and out of a specific sector over a given period. This data, collected from the website (https://www.joinquant.com/, accessed on 1 May 2023), categorizes fund flows into small, medium, large, and extra-large orders based on the transaction amount. Notably, the main fund comprises large and extra-large orders. Large orders are transactions involving more than one hundred thousand shares or CNY two hundred thousand, but less than one hundred thousand shares or CNY one million. Extra-large orders are those involving more than five hundred thousand shares or CNY one million. The capital flow data $CF_{i}\left(t\right)$ represents orders of more than one hundred thousand shares or CNY two hundred thousand. The data indicate the transaction amount of orders that meet the above criteria. A positive amount signifies inflow, a negative amount indicates outflow, and otherwise, it is 0.

To supplement the capital flow data, the daily closing prices for the same stocks during the same period were also obtained using the WIND terminal. Stocks with missing data for more than 200 consecutive trading days within the specified time frame were omitted from the analysis. Ultimately, 152 individual stocks are included in this study.

Herein, we denote the main capital flow and closing price of the *i*-th stock on day *t* as $Y_{i}\left(t\right)$, respectively. Additionally, the price return is defined as follows:(1)$$\begin{array}{c} R_{i}\left(t\right)=\left(Y_{i}\left(t\right)-Y_{i}\left(t-1\right)\right)/Y_{i}\left(t-1\right). \end{array}$$
TE, a significant tool in the information theory, serves as a measure of uncertainty or unpredictability of a discrete variable. It quantifies the average information about the forthcoming state of the destination variable *X*, present in the source variable *Y* but not contained in the historical states of *X*. We presume that variable *X* is influenced by the *k* previous states of *X* itself and the *l* previous states of variable *Y*. Here we take k=l=1 for simplicity. Accordingly, TE from time series $\{y_{t}$} to $\left\{x_{t}\right\}$ can be defined as follows: (2)$$TE_{y_{t}\to x_{t}}=\sum_{x_{t+1},x_{t},y_{t}}p\left(x_{t+1},x_{t},y_{t}\right)\log_{2}\frac{p(x_{t+1}|x_{t},y_{t})}{p(x_{t+1}|x_{t})},$$
where p(x) represents probability and the summations correspond to the sum over the discrete states *N* for the time series when we discretize the time series. $p(x_{t+1},x_{t},y_{t})$ is the joint probability of $x_{t+1}$, $x_{t}$, and $y_{t}$. $p(x_{t+1}|x_{t},y_{t})$ is the conditional probability of variable $x_{t+1}$ providing $x_{t}$ and $y_{t}$. By substituting X and Y with the main capital flow or price return time series, we obtain $TE_{ij}$ for capital flows or price returns of stock *i* and *j* [40,41,42,43], i.e.,
(3)$$\begin{array}{c} TE_{ij}=\sum_{CF_{j}\left(t+1\right),CF_{j}\left(t\right),CF_{i}\left(t\right)}p\left(CF_{j}\left(t+1\right),CF_{j}\left(t\right),CF_{i}\left(t\right)\right)\log_{2}\frac{p(CF_{j}\left(t+1\right)|CF_{j}\left(t\right),CF_{i}\left(t\right))}{p(CF_{j}\left(t+1\right)|CF_{j}\left(t\right))}, \end{array}$$
(4)$$\begin{array}{c} TE_{ij}^{pri}=\sum_{R_{j}\left(t+1\right),R_{j}\left(t\right),R_{i}\left(t\right)}p\left(R_{j}\left(t+1\right),R_{j}\left(t\right),R_{i}\left(t\right)\right)\log_{2}\frac{p(R_{j}\left(t+1\right)|R_{j}\left(t\right),R_{i}\left(t\right))}{p(R_{j}\left(t+1\right)|R_{j}\left(t\right))}, \end{array}$$
Here We use N=50 to calculate TE, varying $N_{x}=N_{y}$ from 2 to 80 in calculations between capital flows [13]. We discover that TE does not increase when *N* surpasses 50.

To extract the strength and direction of information flow between different capital flow time series, we calculate $TE_{ij}$ between stock *i* and stock *j* with transfer entropy method. Thus, TE network of cash flow transfer network is constructed.

Taking an average over stocks in business sectors, we obtain the transfer entropy between business sectors $TE_{ij}^{sector}$. Denoting the inflow and outflow transfer entropy for each business sectors as follows,
(5)$$\begin{array}{c} TE_{i}^{in}=\frac{1}{m}\sum_{i=1}^{m}TE_{ji}^{sector}. \end{array}$$
(6)$$\begin{array}{c} TE_{i}^{out}=\frac{1}{m}\sum_{j=1}^{m}TE_{ij}^{sector}. \end{array}$$
The outflow $TE_{i}^{out}$ can be regarded as the influence of a sector in the market, and the inflow $TE_{i}^{in}$ can be regarded as attention level from main funds.

To investigate the spreading dynamics of the transfer networks and explore the significance and traveling time of each node within complex financial networks, we implemented a Monte Carlo simulation of the SIS model on the capital flow transfer network [37,38]. In the SIS infectious disease model, each node exhibits two distinct states: susceptible (S) and infected (I) [44,45,46,47]. The dynamic equation of this model is defined as [37]:(7)$$\frac{dx_{i}\left(t\right)}{dt}=-x_{i}\left(t\right)+\sum_{j=1}^{N}A_{ji}\left(1-x_{i}\left(t\right)\right)x_{j}\left(t\right).$$
Here, $x_{i}\left(t\right)$ represents the probability of infection for node *i*, suggesting the likelihood of node *i* being exposed to a perturbation. The first term on the right side of the equation denotes the recovery process of node *i*, reflecting the tendency of node *i* to return to its original state. Conversely, the second term symbolizes the infection process, illustrating the influence of neighboring node *j* on node *i*. Analogically, a stock is infected if the stock is exposed to risk (information), and the remaining stocks that have not been exposed to risk (information) yet are susceptible. The matrix $A_{ji}$ captures the impact of node *j* on node *i*, encoding the connectivity pattern and the strength of influence between nodes in the network. To explore the spreading dynamics of the capital flow transfer networks, evaluating the importance and transit time of each node, we define $A_{ij}=TE_{ij}^{sector}$ in our paper.

To explore the spreading dynamics within the transfer network, we employ a perturbation $\Delta x_{j}$ on node *j*, and observe its propagation across the network. This method allows us to determine the traveling time, denoted as $T_{ij}$:(8)$$\Delta x_{i}\left(t=T_{ji}\right)=\eta \Delta x_{i}\left(t\to \infty \right).$$
Herein, we use the variable $T_{ij}$ to denote the time at which node *i* reaches an η-fraction (in this case, η=0.5) of its final response to the perturbation originating from its neighboring node *j* [38]. For simplicity, we consider t→∞ as the time when stock *i* reaches its ultimate response to the perturbation from node *j*. Therefore, $T_{ij}$ represents the half-life of stock *i*’s response to the perturbation. We measured the signal propagation time between any two sectors ij. If *i* cannot propagate to *j* and we set the propagation time to positive infinity, then its reciprocal is 0.

The network eventually stabilizes. To measure the importance of node *i* or edge ij during the perturbation’s spreading process originating from source *n*, we halt the dynamic flow through *i* or ij. We define the dynamic flow through *i* or ij from node *n* as follows [37]:(9)$$\begin{array}{c} F_{n}^{i}=1-\frac{\sum_{n=1,i}^{N}\left|\frac{dx_{m}/x_{m}}{dx_{n}/x_{n}}\right|}{\sum_{n=1}^{N}\left|\frac{dx_{m}/x_{m}}{dx_{n}/x_{n}}\right|}, \end{array}$$(10)$$\begin{array}{c} F_{n}^{ij}=1-\frac{\sum_{n=1,ij}^{N}\left|\frac{dx_{m}/x_{m}}{dx_{n}/x_{n}}\right|}{\sum_{n=1}^{N}\left|\frac{dx_{m}/x_{m}}{dx_{n}/x_{n}}\right|}, \end{array}$$
where $\frac{dx_{m}/x_{m}}{dx_{n}/x_{n}}|$ signifies the influence of the local perturbation from source *n* on node *m* in the stable state [48,49]. Here $\sum_{n=1}^{N}$ means a sum over *n* and $\sum_{n=1,i}^{N}$ means a sum over *n* when *i* node is frozen. similarly, $\sum_{n=1,ij}^{N}$ means a sum over *n* when edge ij is frozen. Freezing node *i* means to artificially set the node unperturbed and hence preventing it from propagating the signal onward. Likewise freezing the link ij means to artificially block the pathway thus set the signal can not spread from node *i* to node *j*. By averaging over different nodes *n*, we derive the dynamic flow of node *i*, and the dynamic flow $F_{ij}$ from node *i* to node *j*:(11)$$\begin{array}{c} F_{i}=\frac{1}{N}\sum_{n=1}^{N}F_{n}^{i}. \end{array}$$(12)$$\begin{array}{c} F_{ij}=\frac{1}{N}\sum_{n=1}^{N}F_{n}^{ij}. \end{array}$$

## 3. Results

### 3.1. Inflow and Outflow of Transfer Entropy between Business Sectors

The Chinese stock market, an emerging market, exemplifies a complex financial network. The essence of finance involves the exchange of capital, in which information plays a crucial role. The temporal evolution of the main capital flow is presented in Figure 1a. Figure 1a shows substantial trading activity in the capital flow market during 2015. By examining the absolute values of the main funds each day as depicted with the solid line in Figure 1a, it becomes apparent that the volume of main funds traded from 2018 to 2022 surpassed that of 2010 to 2014. Simultaneously, it can be observed that the annual average of main funds is negative. This suggests that within the primary market, there typically exists a pattern of buying in smaller to medium orders, while larger orders are being sold off. Notably, the outflow of capital peaked in 2015 due to the aftermath of the financial crisis. The impact of the COVID-19 pandemic led to a significant outflow of capital in 2021.

To extract the strength and direction of information flow between different capital flow time series, we calculate $TE_{ij}$ between stock *i* and stock *j* with transfer entropy method and directed the maximum plane filtering (PMFG). TE network of cash flow transfer network is displayed in Figure 1b. We can observe that stocks with close connections often belong to different sectors.

Our dataset consists of 152 individual stocks covering 25 business sectors. We removed sectors with only one or two stocks, leaving 21 business sectors. The sector names and their respective ticks are detailed in Table 1.

Taking an average over stocks in business sectors, we obtain the transfer entropy between business sectors $TE_{ij}^{sector}$, where the outflow $TE_{i}^{out}$ can be regarded as the influence of a sector in the market, and the inflow $TE_{i}^{in}$ can be regarded as attention level from main funds. Ranking by $TE_{i}^{in}$ and $TE_{i}^{out}$, we list the top three business sectors in Table 2.

Analysis of the results unveils intriguing patterns concerning the inflow and outflow TEs ($TE_{i}^{in}$ and $TE_{i}^{out}$) of the top three business sectors. Intriguingly, these TEs show symmetric behavior. However, the highest-ranked sectors have experienced changes over time. Specifically, the food and drink sector, which mainly consists of liquor stocks within the HS300 index, frequently ranks among the top three. The inflow TE of the food and drink sector has made the top three on eight occasions, while the outflow TE has reached this rank nine times. This trend suggests the significant standing of the drink industry in the Chinese market.

The automobile sector has shown an impressive trend, with its outflow and inflow TEs ranking first in 2014 and 2017, respectively. This pattern suggests the sector’s unique dominance during these periods. The utilities sector, primarily comprising electricity and gas companies, did not rank in the top three before 2018. However, in 2022, its inflow TE surpassed all others to claim the first spot. Conversely, its outflow TE has not yet ranked in the top three. Much like the automobile sector in 2014, the utilities sector is expected to emerge as one of the focuses of market in the coming few years, partly due to the current turbulent world situation.

We sort sectors by value of transfer entropy, and Figure 2 illustrates the ranking of $TE_{i}^{in}$ and $TE_{i}^{out}$ for four typical sectors. Figure 2a displays a notable rise in the rank of the petrochemicals sector in 2018, coupled with a higher inflow TE compared to its outflow TE. In 2018, U.S. sanctions on Iran disrupted the crude oil supply market. Concurrently, the escalating trade disputes dampened oil demand forecasts, and the continuous surge in U.S. shale oil production intensified concerns over an oil surplus. Such a trend underscores that, starting from 2018, the petrochemicals sector commenced garnering significant interest from main funds, a probable prelude to the gradual rebound of oil prices thereafter.

The year 2014 was a pivotal year for the tech industry. Figure 2b demonstrates a marked elevation in the rank of the computer sector. This surge coincided with significant events like the Alibaba Group’s official listing on the New York Stock Exchange. Additionally, the same year witnessed the establishment of private banks, financed by private capital firms including Tencent and Alibaba, obtaining approval from China’s banking regulatory commission, marking the commencement of financial reforms. From 2019 onwards, the computer sector’s inflow transfer entropy (TE) consistently surpassed its outflow TE, suggesting sustained growth and development from 2019 to 2022. Nevertheless, the sector’s overarching influence in the market appears to have diminished in the ensuing years.

Figure 2c delineates that between 2012 and 2017, the automobile sector’s ranking was relatively high, denoting a phase of robust development. This period witnessed the sector’s pronounced influence in the market, with significant investments from main funds. The backdrop to this growth can be attributed to the Sino–Japanese relations deteriorating in 2012, resulting in a sharp decline in the market share of Japanese cars. Concurrently, the evolution of new energy vehicles marked the onset of a golden era for China’s electric vehicle industry. The industry’s growth aligned with the priorities set in China’s 13th Five-Year Plan. By 2017, the country’s automobile production and sales soared to 29.015 million and 28.879 million vehicles, respectively, making China the global leader in the automobile market.

Between 2011 and 2020, the food and drink sector in China saw highs and lows, experiencing peaks, transitions, adjustments, and subsequent revivals. Specifically, during 2011–2012, the food and drink sector soared to its zenith. However, a deep adjustment period followed from 2013 to mid-2016. By the latter half of 2016 through 2019, the sector witnessed a structural revival. The advent of the COVID-19 pandemic in 2020 delivered a direct market shock to the food and drink sector. Mainstream enterprises in this sector, after a brief period of uncertainty, swiftly recalibrated their strategies. This trajectory aligns consistently with the changes in transfer entropy rankings illustrated in the figures. Figure 2d emphasizes the food and drink sector’s dominance in the Chinese market since 2011. Yet, from 2021 onwards, the inflow TE began trailing the outflow TE, suggesting a potential deceleration in capital inflows and a waning influence on the Chinese market.

Overall, analyzing the sector rankings provides insights into the shifting dynamics of capital flows and the changing levels of attention and impact each sector receives.

### 3.2. Community Structure of the Transfer Network

The presence of a community structure is a significant attribute of complex financial networks, representing a set of nodes interconnected by dense edges. To further understand this, we delve into the community structure within the capital flow transfer network. This study builds the capital flow transfer network based on TE measurements between individual stocks. We utilize community detection techniques to identify communities within the capital transfer network and analyze the temporal evolution of the community structure from 2010 to 2022.

A fully connected matrix cannot form communities.To filter symmetric networks, we utilize the maximum plane filtering (PMFG) method. The edges are sorted in descending order, and when inserting edges A and B, we ensure that they adhere to the maximum plane principle, with no edge existing from B to A. This ensures that there is only one edge connecting any two nodes, and the maximum number of edges that can be inserted is 3(N−2), where *N* represents the total number of nodes. Based on the half-life values $T_{ij}$, we generate the PMFG graph denoted as $T_{ij}$ [25,50,51]. Subsequently, we apply the infomap directed network multi-layer clustering method [52] to extract the community structure within the transfer network.

In the previous subsection, we create the filtered transfer network denoted as $TE_{ij}^{\prime }$ [25,50,51]. We then use the infomap directed network multi-layer clustering method [52] to extract the community structure within the transfer network. Figure 3a–e illustrate the community structure of the capital flow transfer network in 2010, 2013, 2016, 2019, and 2022. In 2010, 2013, and 2019, capital flowed from the “first” community to the “non-first” community, while in 2013 and 2022, the trend was reversed, with capital moving from the “non-first” community to the “first” community. By analyzing the average volume of main funds within stocks of both the “first” and “non-first” communities, we observed that the transfer of entropy information consistently flows from the community with a higher average main fund to the one with a lower average. By observing the stock characteristics within the community, it can be found that communities are not formed according to business sectors, such as the “return rate” community. Each community includes most sectors, because institutional investors tend not to put all their eggs in one basket (sector). For comparison, the community structure of TE of price returns is depicted in Figure 3f–j. The community structure of TE of price returns is entirely for the inflow of the largest community, markedly differing from the community structure of the main funds. In 2019, maybe due to the China-U.S. trade war, traditional investments began to lose their significance. This triggered a transformation in the Chinese economy, which was reflected in the increase in the number of communities in 2019 compared to other years.

### 3.3. Spreading Dynamics of the Transfer Network

Relying on the TE between business sectors $TE_{ij}^{sector}$, we delve into the spreading dynamics of the capital flow transfer networks, evaluating the importance and transit time of each node. Defining $A_{ij}=TE_{ij}^{sector}$, the dynamic flow through all nodes and edges, denoted as $F_{i}$ and $F_{ij}$, are calculated. A larger dynamic flow of a node (edge) indicates a greater significance of the corresponding sector (the interaction between two sectors) in the market’s information dissemination. Additionally, the traveling time from the *i*-th sector to *j*-th sector, $T_{ij}$, is calculated. The average traveling time from the *i*-th sector to other sectors, denoted as $T_{i}$, is obtained by averaging over *j*. A shorter traveling time suggests a faster signal spread from the *i*-th sector to the *j*-th sector, and likewise, to other sectors.

The results are illustrated in Figure 4. As depicted in Figure 4a, the reciprocal of average traveling time $1/T_{i}$ from business sectors—food and drink, coal, and public utility—rank first to third, respectively. Figure 4b shows that the dynamic flow $F_{i}$ of the steel, real estate, and mechanical equipment sectors are the top three. As indicated in Figure 4c, the traveling time $T_{i}$ from the business sectors—food and drink, coal, and public utility—to other sectors is the shortest, with the strongest dynamic flow $F_{ij}$ between these sectors, as depicted in Figure 4d.

The temporal evolution of dynamic flows and traveling time is shown in Figure 5. As seen in Figure 5a, the reciprocal of average traveling time $1/T_{i}$ from the business sector medical biology remains high from 2013 to 2018, and $1/T_{i}$ from the food and drink sector ranks at the forefront from 2016 to 2022. This indicates a rapid signal spread from these two sectors to others. As Figure 5b reveals, $F_{i}$ of the steel sector remains high from 2014 to 2017 and from 2019 to 2022, denoting its influence in the capital flow transfer network during these periods.

## 4. Discussion

The analysis of the financial system as a complex network offers valuable insights into capital exchange dynamics. By employing the TE method, we have successfully extracted the strength and direction of information flows among different capital flow time series. This enables us to scrutinize the community structure of the transfer networks. Furthermore, our examination of the spreading dynamics of capital flow transfer networks facilitates the exploration of the importance and traveling time of each node.

Our analysis yields several significant observations. Firstly, we note that the drink industry maintains a substantial role within the Chinese financial market, indicative of its overarching prominence. Secondly, we observe a growing influence of the computer industry, which has been gaining attention since 2014. Moreover, we identify that the community structure of the capital flow transfer network significantly deviates from the network built exclusively on stock prices. It is particularly noteworthy that the largest sector primarily comprises industry leaders who attract significant investments from main funds.

In our in-depth exploration of the reciprocal of the average traveling time from sectors such as food and drink, coal, and public utility to other sectors, it is the highest. Concurrently, the dynamic flow ($F_{ij}$) between these sectors displays significant strength. Looking at specific sectors, the dynamic flow of the steel, real estate, and Mechanical equipment sectors ranks within the top three. Further, the reciprocal of the average traveling time from the medical biology sector stays consistently high from 2013 to 2018, while the food and drink sector leads from 2016 to 2022. These findings suggest that signals disperse rapidly from these sectors to others. Additionally, the steel sector exhibits consistent influence within the capital flow transfer network during certain periods, specifically from 2014 to 2017 and 2019 to 2022.

Overall, our findings offer valuable insights into the mechanisms governing capital flows within complex financial networks. We successfully identify nodes of paramount importance in the capital propagation process and nodes that expedite the dissemination of signals to other nodes.

## Figures and Tables

**Figure 1 entropy-25-01240-f001:**
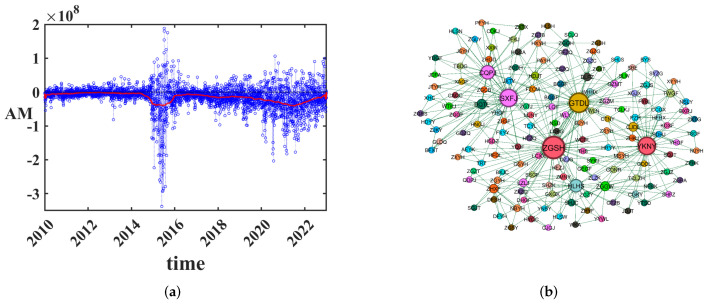
(**a**) Time evolution of average main capital (AM). The daily average main capital (AM) and annual average main capital is displayed in blue dotted line and red solid line, respectively. (**b**) TE network of cash flow transfer network. There are 152 individual stocks, the abbreviations of which are marked. Different colors represent different sectors.

**Figure 2 entropy-25-01240-f002:**
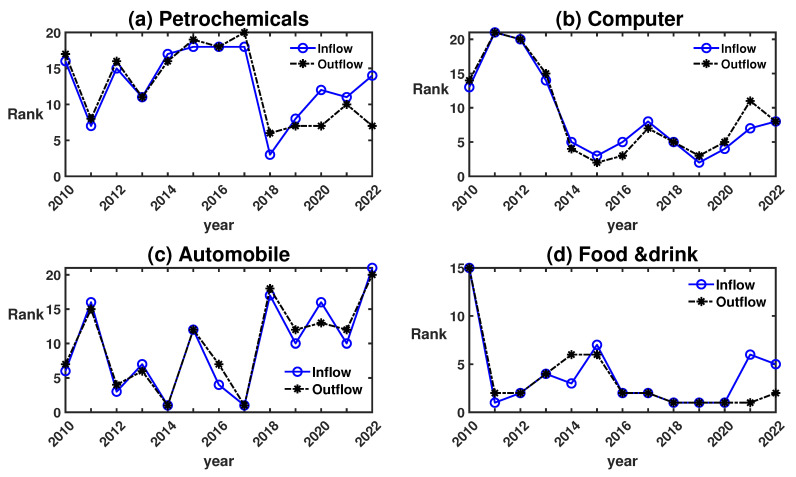
The transfer entropy rank of petrochemicals, computer, automobile, and food and drink sectors.

**Figure 3 entropy-25-01240-f003:**
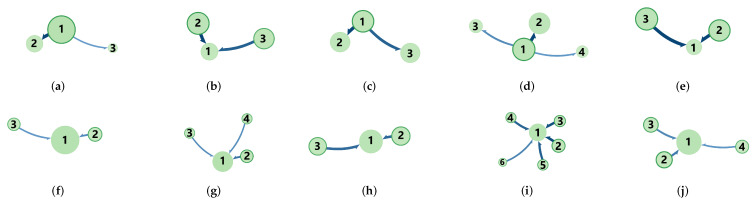
(**a**–**e**) Community structure of capital flow transfer network in different years. (**f**–**j**) Community structure of transfer network based on price returns in different years. The size of nodes and the width of edges represent the number of nodes in the community and the capital transfer flow between two communities, respectively.

**Figure 4 entropy-25-01240-f004:**
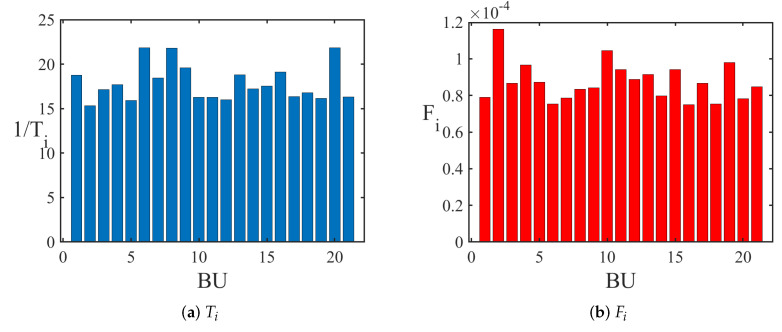
(**a**) The reciprocal of average traveling time for different business sectors. (**b**) Dynamic flow of different business sectors. (**c**) The reciprocal of traveling time between different business sectors. The depth of the color represents the size of the value of $1/T_{ij}$. (**d**) Dynamic flow between different business sectors. The depth of the color represents the size of the value of $F_{ij}$.

**Figure 5 entropy-25-01240-f005:**
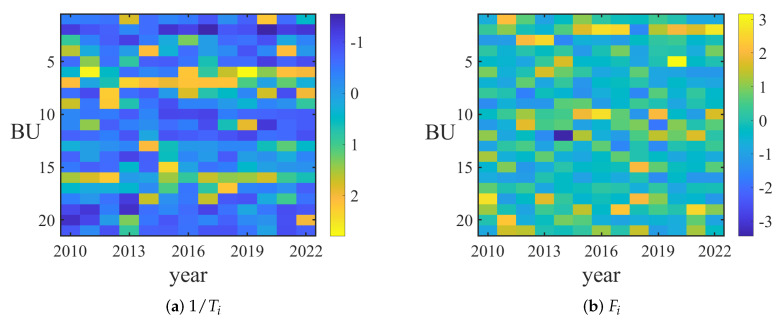
(**a**) The reciprocal of average traveling time for different business sectors form 2010 to 2022. The depth of the color represents the size of the value of $1/T_{i}$. (**b**) Dynamic flow of different business sectors form 2010 to 2022. The depth of the color represents the size of the value of $F_{i}$. For comparison, we normalize $1/T_{i}$ and $F_{i}$ for different sectors.

**Table 1 entropy-25-01240-t001:** The sector names and code, and the number of stocks in sector.

Code	Name	Short-Name	Stock-Num
801030	Chemical	Chemical	5
801040	Steel	Steel	3
801050	Non-ferrous metals	NFM	6
801080	Electronic	Electronic	9
801110	Household appliances	HA	4
801120	Food–drink	Food–drink	10
801150	Medicinal organisms	MO	13
801160	Utilities	Utilities	6
801170	Transportation	Transportation	9
801180	Real estate	RE	5
801710	Building materials	BM	5
801720	Building decoration	BD	5
801730	Electrical equipment	EE	8
801740	National defense	ND	7
801750	Computer	Computer	8
801780	Bank	Bank	14
801790	Non-bank financial	NBF	9
801880	Automobile	Automobile	6
801890	Mechanical equipment	ME	5
801950	Coal	Coal	4
801960	Petrochemicals	Petrochemicals	5

**Table 2 entropy-25-01240-t002:** The average inflow and outflow transfer entropy of top three sectors from 2010 to 2022.

Year	First-in	Second-in	Third-in	First-out	Second-out	Third-out
2010	BM	MO	ME	BM	MO	ME
2011	Food–drink	Bank	ME	Bank	Food–drink	ND
2012	Bank	Food–drink	Automobile	Bank	Food–drink	BM
2013	Chemical	HA	MO	Chemical	HA	MO
2014	Automobile	MO	Food–drink	Automobile	HA	MO
2015	ND	NBF	Computer	ND	Computer	NBF
2016	Chemical	Food–drink	MO	Chemical	Food–drink	Computer
2017	Automobile	Food–drink	NBF	Automobile	Food–drink	HA
2018	Food–drink	Utilities	Petrochemicals	Food–drink	NBF	EE
2019	Food–drink	Computer	RE	Food–drink	MO	Computer
2020	Food–drink	Utilities	HA	Food–drink	HA	Electronic
2021	EE	HA	Electronic	Food–drink	NFM	EE
2022	Utilities	Coal	Chemical	Coal	Food–drink	Chemical

## Data Availability

The data used in this study are available from the authors.

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
