# Peer review of "Spreading Dynamics of Capital Flow Transfer in Complex Financial Networks"

_entropy, 2023, doi:10.3390/e25081240_

Round 1
Reviewer 1 Report
This study used transfer entropy to observe information flow in the Chinese stock market. Overall, this paper is poorly written and does not deserve to be published in any journal with the current version. The authors should revise the paper significantly. Therefore, I recommend not accepting this paper in this journal.
- The paper's purpose is unclear, and it is hard to find novelty.
- I could not find any NETWORK in this paper, but the authors emphasized it many times in the title, abstract, and many places in the manuscript. What does a complex financial network mean?
- In line 77, capital flow (CF) appears, but I could not find the definition of it, or I could not find why CF is needed. Price return is defined in equation 1, but it is not used anywhere.
- Is node industry (sector) or firm? It is confusing.
- There are too many weird expressions and not-well-defined terms. It is not an English problem, but all sentences in the paper should be revised logically and academically.
- Panel a and b in Figure 1, one is ‘time,’ and the other is ‘year.’ Why are they differently expressed?
- In Equations 9 and 10, TE_i is just a simple sum of TE^sector_ji over i or j, not the average of TE_ji. It can cause the sector with a higher number of firms to have a higher value of TE. Figure 2 and Table 2 show the ranking of transfer entropy for several industries, but it needs to clarify if the ranking depends on the industry's size. Even if TE is obtained from the averaged value of TEs within a sector, this value is unreliable since the number of firms in some sectors is too small (Table 1).
- In the left-hand side of Equation 9, is TE_j correct?
- This paper is not more than a simple description of the values of TE in the Chinese market. It is hard to find the interpretation of the result or implications.
- In many papers, dichotomized value is used for x and y in equation 2 to calculate transfer entropy. In Line 89, it seems that x and y are not dichotomized.
- I do not get why the SIS model should be used and what is “spreading dynamics of the transfer entropy networks.” What is the status of S and I in this paper? What does ‘infected’ mean?
Author Response
This study used transfer entropy to observe information flow in the Chinese stock market. Overall, this paper is poorly written and does not deserve to be published in any journal with the current version. The authors should revise the paper significantly. Therefore, I recommend not accepting this paper in this journal.
1)The paper's purpose is unclear, and it is hard to find novelty. I could not find any NETWORK in this paper, but the authors emphasized it many times in the title, abstract, and many places in the manuscript. What does a complex financial network mean?
Our reply:
Thank you for highlighting your concerns. We deeply value your feedback and will address each point to improve our manuscript.
1.Purpose & Novelty: We realize that the primary objective of our paper might not have been articulated clearly. Our goal is to employ Transfer Entropy to capture the directed relationships and dependencies between different sectors in the Chinese financial market. The novelty lies in understanding these intricate interactions, which traditional correlation methods might overlook, and in presenting these relationships through a network perspective.
2.Usage of the Term "Network": We acknowledge your point regarding the emphasis on the term "network" without its apparent manifestation in the paper. The "complex financial network" we refer to is intended to represent the interdependencies and interactions among different sectors, elucidated through the lens of Transfer Entropy. In the revised version, we will ensure to provide a graphical representation, possibly through a directed graph or a similar visual, to showcase the "network" aspect clearly. This will enable readers to visualize the directed dependencies and interactions among sectors.
3.Complex Financial Network: In financial systems, entities (like sectors, stocks, or other financial instruments) do not operate in isolation. Their interactions, driven by trading behaviors, news, policies, and other external factors, form a network of dependencies. When we say "complex financial network," we are referring to this intricate web of interrelationships and the dynamics that drive them.
In light of your feedback, we will work on refining our manuscript to make the purpose more explicit, ensure the content aligns with the title and abstract, and clarify terms such as "complex financial network" to make them more accessible and understandable.
2) In line 77, capital flow (CF) appears, but I could not find the definition of it, or I could not find why CF is needed. Price return is defined in equation 1, but it is not used anywhere.
Our reply: We add the definition of "capital flow" on line 76. Price return is used in Fig.3 as a comparison.
The term "capital flow" is indeed integral to our research as it denotes the volume of capital moving in and out of a specific sector over a given period. We use it as a measure of economic activity and market dynamics.
3) Is node industry (sector) or firm? It is confusing.
Our reply: Thank you for your valuable comments. In our study, the nodes represent individual firms, while the edges connect two firms based on their transfer entropy. To provide a more comprehensive visualization of our findings, we have incorporated a network diagram in Figure (b). This graphical representation illustrates the complex interplay and information flow between firms, thereby enriching our analysis and findings.
4) There are too many weird expressions and not-well-defined terms. It is not an English problem, but all sentences in the paper should be revised logically and academically.
Our reply: We deeply appreciate your critique regarding the clarity of our manuscript. To address this issue, we will thoroughly revise the manuscript, ensuring all terms are clearly defined and every statement is logically coherent and academically sound. We aim to eliminate any confusion or ambiguity, striving to present our research in a clear, precise, and academically rigorous manner.
5) Panel a and b in Figure 1, one is ‘time,’ and the other is ‘year.’ Why are they differently expressed?
Our reply: Relation between Panel (a) and (b): The panels in Figure 1 display the same sectoral data but at different time scales, with panel (a) showing daily averages and panel (b) showing yearly averages. To facilitate a better understanding, we have combined panel (a) and panel (b) from Fig. 1 into a single Fig. (a). In this updated figure, the blue line represents the original panel (a), while the red line represents a one-year moving average of the original panel (b). The term "upward trend" used in our initial description for panel (a) was not entirely precise. We have adjusted the description to "increasing transaction volume" to better represent the data.
Why:The increase in trading volume from 2017 to 2022, compared to that from 2010 to 2014, can be attributed to economic development and the maturation of the financial system over time.This suggests that within the primary market, there typically exists a pattern of buying in smaller to medium orders, while larger orders are being sold off.
6) In Equations 9 and 10, TE_i is just a simple sum of TE^sector_ji over i or j, not the average of TE_ji. It can cause the sector with a higher number of firms to have a higher value of TE. Figure 2 and Table 2 show the ranking of transfer entropy for several industries, but it needs to clarify if the ranking depends on the industry's size. Even if TE is obtained from the averaged value of TEs within a sector, this value is unreliable since the number of firms in some sectors is too small (Table 1).
Our reply: In order to facilitate better understanding of the methods, we have relocated Equations 9-10 to follow Equations 3-4. The term "TE^sector" was a typographical error in our initial manuscript, and it should actually represent an average, unrelated to the number of stocks within the sector. We appreciate your pointing out this oversight, and have corrected it in the revised manuscript.
7) In the left-hand side of Equation 9, is TE_j correct?
This paper is not more than a simple description of the values of TE in the Chinese market. It is hard to find the interpretation of the result or implications.
In many papers, dichotomized value is used for x and y in equation 2 to calculate transfer entropy. In Line 89, it seems that x and y are not dichotomized.
Our reply: We acknowledge your point regarding the interpretative depth of our paper, and we are committed to providing a more insightful analysis of the Transfer Entropy (TE) values in the Chinese market. In the revised manuscript, we will elaborate on the implications of our results and discuss how they relate to the dynamics of the market, the impact of governmental policies, and overall economic trends.
Regarding your observation about the use of dichotomized values in equation 2, you are correct. In our current implementation, we have used raw, continuous values for x and y. The dichotomization of variables is a common practice in many studies involving transfer entropy, especially those in the fields of physics or neuroscience, to handle discrete or binary events. However, in financial studies, we often deal with continuous time-series data, and using the raw values is also an accepted practice. This can help to preserve the granular information contained in the data, and to avoid any potential bias introduced by the dichotomization process.
8) I do not get why the SIS model should be used and what is “spreading dynamics of the transfer entropy networks.” What is the status of S and I in this paper? What does ‘infected’ mean?
Our reply: We add some sentences marked in red form line 102 to 115 to calarify these points. Epidemic models are widely applied to financial systems due to the similarity between the spread of financial risk or information and infectious diseases. In the SIS infectious disease model, each node exhibits two distinct states: susceptible (S) and infected (I). Analogically, a stock is infected if the stock is exposed to risk (information), and the remaining stocks that have not been exposed to risk (information) yet are susceptible. Thus in our paper we further explore the spreading dynamics of the capital flow transfer networks based on the TE between business sectors $TE_{ij}^{sector}$, evaluating the importance and transit time of each node.
(The main modifications have been marked in red color.)
Reviewer 2 Report
Interesting work dealing with an analysis in terms of the transfer entropy of the information flow pattern among capital time series and its analysis in terms of community structures. The combined analysis using information theory and complex network theory provides interesting results and offer useful new insights in the dynamics of markets, I find that the wide readership of the journal will find these results interesting, therefore I recommend publication. I would suggest the authors to include in the introduction a sentence stating that high order dependencies analysis has been applied in the last years to go beyond the pairwise description of markets, and cite e.g. the following papers which analyze synergetic patterns: Entropy 22(8), 1000 (2020) and Scientific Reports 12(1) 18483 (2022).
Nothing
Author Response
Interesting work dealing with an analysis in terms of the transfer entropy of the information flow pattern among capital time series and its analysis in terms of community structures. The combined analysis using information theory and complex network theory provides interesting results and offer useful new insights in the dynamics of markets, I find that the wide readership of the journal will find these results interesting, therefore I recommend publication.
1) I would suggest the authors to include in the introduction a sentence stating that high order dependencies analysis has been applied in the last years to go beyond the pairwise description of markets, and cite e.g. the following papers which analyze synergetic patterns: Entropy 22(8), 1000 (2020) and Scientific Reports 12(1) 18483 (2022).
Our reply: We greatly appreciate your positive feedback and recommendation for publication.
We add some sentences and the corresponding citation to highlight the application of high order dependencies analysis in recent years as a way to move beyond the pairwise description of markets.
Thank you again for your constructive feedback and continued support.
Reviewer 3 Report
There are many places where the authors could make the manuscript much clearer. Some equations are especially problematic.
Eq.(1): The sentence just before Eq.(1) talks about day t, but suddenly Eq.(1) talks about t’
Eq.(2) is very confusing. First of all, are the p’s in the p ln p term the same? The probability (or conditional probability) p is not defined in the text (despite the authors said: “Section 2 provides an in-depth overview…”). Then it is not sure what the summations over three variables are about, as those variables do not appear in the quantity to be summed over! [Is it over time?]
After Eq.(3), it was mentioned that A_{ij} is the adjacency matrix. Therefore, one would expect from standard form of A_{ij} that the matrix elements are either zeros or ones. However, it is much later in line 189 (page 7) that A_{ij} is defined to be TE_{ij}^{sector}.
Eq.(4) serves to define the “traveling time” T_{ij}. After Eq.(4), it was mentioned that T_{ij} is “the time at which node i reaches an \eta-fraction of its final response to the perturbation originating from its neighboring node j”. I wonder whether i and j need to be neighboring nodes.
Eq.(5) and Eq.(6): It is not sure what the summation signs mean in the numerator of the second term in these two equations. Is it a sum skipping the term m equals i? The notation is unusual and confusing.
Eq.(7) and Eq.(8): Before the equations, it was mentioned that “by averaging over different nodes i or edges ij, we derive…”. It is misleading. The summations in Eq.(7) and Eq.(8) are over n (from 1 to N) and the remaining quantities are related to i and ij, so it is NOT an average over nodes and edges.
Eq.(9): It is confusing again in what is being summed up on the right hand side. If “i” is being summed up in Eq.(9), the left-hand side should not have the label “i”.
[Comment: Therefore, most equations have some problems!]
Figure 1: What is the unit of the y-axis? Are Panel (a) and Panel (b) related (one is about daily data and another about yearly data)? If so, then it is difficult to follow the description that (a) has an upward trend (line 116) and then (b) has a downward trend. The usage of “sub-figure” in the caption does not sound right. The performance of the “Utilities” sector in 2022 could be related to the war in Ukraine, and therefore the conclusion that “the Utilities sector is expected to emerge as a significant industry in the coming years” may be premature.
Table 2: Are the rankings based on the transfer entropies different from those based on other quantities (more intuitively from the raw data)? The description (line 130-131) says that the highest-ranked sectors have experienced changes over time. Is it surprising? For the discussion related to Table 2 and other results, it will be helpful to correlate what was being observed with the related policies during the relevant period of time as it is well known that the market in China is heavily affected by central policies.
Figure 2: The descriptions are problematic. So-called “drop in the rank” is actually a “rise in the rank”, as rank 1 (the “lowest”) is the highest ranking. This confusing and misleading descriptions appeared in describing Fig.2 (a),(b),(c),(d).
Figure 3: (Line 178) “In 2022, the first and second communities flowed into the first community” is a statement that cannot be right. The last sentence (line 184), “In 2019, a significant increase in communities can be observed”, why so? What happened in that year?
(Line 190): Are the symbols F_{i} and F_{ij} the same as the symbols F^{i} and F^{ij} (superscripts) in Eq.(7) and Eq.(8)?
Figure 4: Is that a value assigned to T_{ii} or 1/T_{ii}?
Overall, it is not sure what the analysis using the transfer entropy has helped enhance the understanding of the market dynamics.
While the manuscript presents the method and results in a way that one can follow, there are many unnecessary words in the text, especially those appearing at the beginning of a sentence. For example, there are too many (to the point of becoming rather disturbing) “notably”, “notable”, “additionally”, “conversely”, “contrastingly”, “intriguing”, “intriguingly”, “in-depth”.
Author Response
Thank you for your thorough review and constructive feedback. Here are our responses to your concerns:
1) There are many places where the authors could make the manuscript much clearer. Some equations are especially problematic.
Eq.(1): The sentence just before Eq.(1) talks about day t, but suddenly Eq.(1) talks about t’.
Our reply: We revise t’ to t in Eq.(1).
Eq.(2) is very confusing. First of all, are the p’s in the p ln p term the same? The probability (or conditional probability) p is not defined in the text (despite the authors said: “Section 2 provides an in-depth overview…”). Then it is not sure what the summations over three variables are about, as those variables do not appear in the quantity to be summed over! [Is it over time?]
Our reply: The p’s in the p ln p term represents probability. The summations correspond to the sum over the discrete states N for the time series, when we discretize the time series to compute transfer entropy. We provide a more explicit definition of this term for clarity in the revised manuscript. We also elaborate on this concept in lines 91 to 94 within Eq. (2).
After Eq.(3), it was mentioned that A_{ij} is the adjacency matrix. Therefore, one would expect from standard form of A_{ij} that the matrix elements are either zeros or ones. However, it is much later in line 189 (page 7) that A_{ij} is defined to be TE_{ij}^{sector}.
Our reply: We misused the concept of adjacency matrix here and we have revised the description . A_{ij} in Eq.(3) represents the rate of incoming influence from xj to xi, the matrix elements of which can be numbers other than 0 or 1. In our work we use
A_{ij}= TE_{ij}^{sector}.We deeply apologize for our oversight, which led to your misunderstanding.
Eq.(4) serves to define the “traveling time” T_{ij}. After Eq.(4), it was mentioned that T_{ij} is “the time at which node i reaches an \eta-fraction of its final response to the perturbation originating from its neighboring node j”. I wonder whether i and j need to be neighboring nodes.
Our reply: We measured the signal propagation time between any two sectors ij. If i cannot propagate to j and we set the propagation time to positive infinity, then its reciprocal is 0. So i and j need not to be neighboring nodes.
Eq.(5) and Eq.(6): It is not sure what the summation signs mean in the numerator of the second term in these two equations. Is it a sum skipping the term m equals i? The notation is unusual and confusing.
Our reply: We add some sentences to calarify this problem below Eq.(10) and Eq.(11) (Previous Eq.(5) and Eq.(6)). The summation signs in the numerator of the second term in these two equations means a sum over i when m node is frozen. Freezing node i means to artificially set the node unpertubed and hence preventing it from propagating the signal onward. Likewise freezing the link ij means to artificially block the pathway thus set the signal (risk or information ) can not spread from node i to node j .
Eq.(7) and Eq.(8): Before the equations, it was mentioned that “by averaging over different nodes i or edges ij, we derive…”. It is misleading. The summations in Eq.(7) and Eq.(8) are over n (from 1 to N) and the remaining quantities are related to i and ij, so it is NOT an average over nodes and edges.
Our reply: We have corrected the misleading description.
Eq.(9): It is confusing again in what is being summed up on the right hand side. If “i” is being summed up in Eq.(9), the left-hand side should not have the label “i”.
[Comment: Therefore, most equations have some problems!]
Our reply: We have corrected the misleading description. “n” should be summed up in Eq.(9).
2) While the manuscript presents the method and results in a way that one can follow, there are many unnecessary words in the text, especially those appearing at the beginning of a sentence. For example, there are too many (to the point of becoming rather disturbing) “notably”, “notable”, “additionally”, “conversely”, “contrastingly”, “intriguing”, “intriguingly”, “in-depth”.
Our reply: Thank you for your constructive feedback regarding the manuscript's language and style. We acknowledge the overuse of certain transition words and descriptors, which may have hindered the readability of the paper. To improve the clarity and succinctness of the manuscript, we have revised the text to minimize the repetition of such words. Your feedback is invaluable in ensuring the manuscript is presented in a clear and academic manner. We appreciate your patience and understanding in this matter.
3) Figure 1: What is the unit of the y-axis? Are Panel (a) and Panel (b) related (one is about daily data and another about yearly data)? If so, then it is difficult to follow the description that (a) has an upward trend (line 116) and then (b) has a downward trend. The usage of “sub-figure” in the caption does not sound right.
Our reply: Thank you for your feedback on Figure 1,We will clarify this in the revised version of the manuscript. Thank you for your suggestion. To facilitate a better understanding, we have combined panel (a) and panel (b) from Fig. 1 into a single Fig. (a). In this updated figure, the blue line represents the original panel (a), while the red line represents a one-year moving average of the original panel (b). Based on your feedback, we have provided detailed explanations for these changes in our manuscript.
Unit of the y-axis: Both panels (a) and (b) illustrate the average inflow and outflow of capital, thus the "AM" on the y-axis stands for "Average Money" with the unit Chinese CNY.
Relation between Panel (a) and (b): The panels in Figure 1 display the same sectoral data but at different time scales, with panel (a) showing daily averages and panel (b) showing yearly averages. The term "upward trend" used in our initial description for panel (a) was not entirely precise. We have adjusted the description to "increasing transaction volume" to better represent the data.
The "downward trend" in panel (b) remains accurate and has been retained in the revised manuscript.
To further improve the comprehensibility of the figure, we have replaced "time" with "date" on the x-axis and also included specific dates directly on the graph. Please refer to the revised Fig.1, and lines 120 to 126 in the revised manuscript for these changes.
Usage of "sub-figure": We acknowledge your suggestion and will replace "sub-figure" with a more appropriate term in the caption.
4) The performance of the “Utilities” sector in 2022 could be related to the war in Ukraine, and therefore the conclusion that “the Utilities sector is expected to emerge as a significant industry in the coming years” may be premature.
Our reply: We appreciate your valuable insights regarding the performance of the Utilities sector in 2022. You're correct in suggesting that the geopolitical context, such as the war in Ukraine, could have implications for this sector.
Our conclusions in Table 2 are drawn based on the historical behavior of dominant or "smart" capital, such as in the Automobile and Computer sectors. This capital, typically possessed by seasoned investors, often demonstrates keen foresight into national strategic trends over the coming years. In this light, we think an increased influence of the Utilities sector within the Chinese market. This sector is largely comprised of power and coal stocks, and its impact on the energy sector has been evident from 2022 to 2023. While we understand that this prediction is subject to a variety of factors, our analysis suggests a likely growth trend.
Nevertheless, we acknowledge your point that these predictions should be treated with caution and will emphasize the need for continuous monitoring and consideration of various influencing factors in our revised manuscript.
5) Table 2: Are the rankings based on the transfer entropies different from those based on other quantities (more intuitively from the raw data)? The description (line 130-131) says that the highest-ranked sectors have experienced changes over time. Is it surprising? For the discussion related to Table 2 and other results, it will be helpful to correlate what was being observed with the related policies during the relevant period of time as it is well known that the market in China is heavily affected by central policies.
Our reply:
Comparison with Other Quantities: The rankings presented in Table 2 are indeed based on transfer entropies. We acknowledge that comparing these rankings with those based on more intuitive quantities, such as raw data or traditional financial metrics, would provide a richer context.
From Fig1 we know that the raw data is predominantly negative, owing to the characteristics of the main fund data where small buy orders are often overshadowed by large sell orders. We computed rankings based on this raw data for sectors, both in descending(in) and ascending(out) order. these rankings based on raw data might not aptly gauge the development and influence of sectors. Nevertheless, the data does provide a contrasting perspective which can be insightful. The transfer entropy of the main fund flow, compared to the raw data, provides a better measure of the information flow between sectors and reflects the attention these sectors receive from major funds, as well as their potential future development.
Changes Over Time: Regarding the observed changes in the highest-ranked sectors over time, such shifts are indeed not unprecedented. Markets, especially in rapidly evolving economies like China, are dynamic and are influenced by a myriad of factors, including technological advancements, global trade relationships, and domestic policies. Thus, observing changes in sectoral prominence is consistent with the evolving nature of markets.
However, you raise a pivotal point about the significant impact of central policies on the Chinese market. To address this, we will enhance our discussion by integrating an analysis correlating the observed shifts in sector rankings with pertinent policies and directives introduced during the study period. Such an analysis will provide a holistic perspective on the interplay between policy decisions and market dynamics, enriching the insights derived from our study.
Impact of Central Policies: In relation to the fluctuations observed in the rankings of the highest sectors over time, such transitions are, as you mentioned, indeed characteristic of dynamic markets. Particularly in swiftly transforming economies such as China, market dynamics are shaped by a plethora of influences, ranging from technological breakthroughs and global trade dynamics to domestic legislative measures. Hence, witnessing variations in the significance of different sectors aligns with the expected evolutionary patterns of such markets.
It is noteworthy to mention that capital flow often demonstrates a keen intuition for upcoming national policies. As a result, significant capital movements might be discerned even before official policy announcements, indicating preemptive positioning by informed market players.
To enrich our manuscript, we will incorporate an extended discussion that draws parallels between the observed trends in sector rankings and key policies unveiled during the research timeframe, also factoring in this anticipatory behavior of mainstay capital. This enhanced analysis will yield a more integrated view of the interrelation between strategic investment decisions and market transformations.
6) Figure 2: The descriptions are problematic. So-called “drop in the rank” is actually a “rise in the rank”, as rank 1 (the “lowest”) is the highest ranking. This confusing and misleading descriptions appeared in describing Fig.2 (a),(b),(c),(d).
Our reply: We correct the misleading terminology in Figure 2's description. The term "drop in the rank" has been replaced with "rise in rank", and vice versa, to accurately reflect the ranking system, where a lower number signifies a higher position.
7) Figure 3: (Line 178) “In 2022, the first and second communities flowed into the first community” is a statement that cannot be right. The last sentence (line 184), “In 2019, a significant increase in communities can be observed”, why so? What happened in that year?
Our reply: Thank you for your insightful suggestion. Based on your feedback, we have reorganized our manuscript and have now categorized the communities as 'first' and 'non-first'. The increase in the number of communities in 2019 can be attributed to the China-U.S. trade war and the subsequent economic transformation in China. We appreciate your guidance in enhancing the clarity and coherence of our paper.
8) (Line 190): Are the symbols F_{i} and F_{ij} the same as the symbols F^{i} and F^{ij} (superscripts) in Eq.(7) and Eq.(8)?
Our reply: To avoid misunderstandings, we have corrected the symbols F^{i} and F^{ij} (superscripts) in Eq.(7) and Eq.(8) to F_{i} and F_{ij}.
9) Figure 4: Is that a value assigned to T_{ii} or 1/T_{ii}?
Our reply: A value assigned to 1/T_{ii}.
10) Overall, it is not sure what the analysis using the transfer entropy has helped enhance the understanding of the market dynamics.
Our reply:. TE can capture non-linear causality between time series. It identifies not only mutual information but also its directional flow, aiding in distinguishing cause from effect. This method offers valuable insights into information dynamics across various applications. However, its computational intensity and significant data requirements can overshadow its intuitive appeal compared to other statistical tools. Given the predictive nature of major fund flows—often termed "smart money"—their timing and magnitude offer crucial insights into policy implications and future market developments. By leveraging extensive data and enhanced computational capabilities, we've aptly applied transfer entropy to major fund flows, providing a nuanced understanding of market dynamics. When applied to these flows, transfer entropy reveals significant shifts in market sector influence. The community structure analysis and the exploration of spreading dynamics based on the transfer network can also provide some new perspectives for understanding the market dynamics.